# Exploration of the Contribution of Fire Carbon Emissions to PM₂.₅ and Their Influencing Factors in Laotian Tropical Rainforests

Zhangwen Su [1] , Zhenhui Xu [1], Lin Lin [2] , Yimin Chen [1], Honghao Hu [1], Shujing Wei [3] and Sisheng Luo [3,*]

1   Zhangzhou Institute of Technology, Zhangzhou 363000, China
2   Earth System Science Interdisciplinary Center, University of Maryland, College Park, MD 20740, USA
3   Guangdong Academy of Forestry, Guangzhou 510520, China
*   Correspondence: luoss2008@sinogaf.cn

**Abstract:** It is of great significance to understand the drivers of PM₂.₅ and fire carbon emission (FCE) and the relationship between them for the prevention, control, and policy formulation of severe PM₂.₅ exposure in areas where biomass burning is a major source. In this study, we considered northern Laos as the area of research, and we utilized space cluster analysis to present the spatial pattern of PM₂.₅ and FCE from 2003–2019. With the use of a random forest and structural equation model, we explored the relationship between PM₂.₅ and FCE and their drivers. The key results during the target period of the study were as follows: (1) the HH (high/high) clusters of PM₂.₅ concentration and FCE were very similar and distributed in the west of the study area; (2) compared with the contribution of climate variables, the contribution of FCE to PM₂.₅ was weak but statistically significant. The standardized coefficients were 0.5 for drought index, 0.32 for diurnal temperature range, and 0.22 for FCE; (3) climate factors are the main drivers of PM₂.₅ and FCE in northern Laos, among which drought and diurnal temperature range are the most influential factors. We believe that, as the heat intensifies driven by climate in tropical rainforests, this exploration and discovery can help regulators and researchers better integrate drought and diurnal temperature range into FCE and PM₂.₅ predictive models in order to develop effective measures to prevent and control air pollution in areas affected by biomass combustion.

**Keywords:** PM₂.₅ drivers; fire carbon emissions; climate factor; spatial distribution; tropical rainforest

## 1. Introduction

PM₂.₅ (equivalent diameter less than 2.5 μm in aerodynamics for particulate matter) has been regarded as the most important air pollutant all over the world during the past two decades, especially in tropical Southeast Asia (SEA) with frequent open-air biomass burning (OBB), which includes wildfires and agricultural burning (i.e., burning of crop residues and land clearing) [1]. About one million wildfires occur worldwide annually, affecting around 3.5 million square kilometers of vegetation along with carbon emissions equivalent to one-third of fossil fuel burning [2]. Nearly 70% of these wildfires occur in tropical forest ecosystems on both sides of the equator. Several studies reported that global carbon emissions caused by wildfires were about 2 PgC/year during 1997–2015, of which the net $CO_2$ emissions from tropical deforestation and peatland incineration were about 0.5 PgC/year, contributing significantly to the increased atmospheric $CO_2$ concentrations [3–5]. To make matters worse, there are about 3.3 million people worldwide dying prematurely from poor air quality; 5% to 8% of these deaths are attributed to air pollution caused by fires, which is the main cause of the increased mortality and environmental threats in tropical regions [4].

The tropics, which have two-thirds of the world's terrestrial biodiversity, have been threatened by disasters, including wildfires, exacerbated by climate change and manmade

disturbances [6]. The Indochina Peninsula in SEA is one of the most important tropical rainforests on Earth. Under the influence of the South Asian monsoon, the abundant precipitation and suitable temperature in the rainy season establish high coverage in the heavily forested region [7]. However, this region is suffering from varying degrees of damage; the rapid development of agriculture has brought about the cultivation of previously forested areas. At the same time, a large amount of biomass combustion also provides an important source of haze pollution in this area [7]. A large number of studies and satellite maps have confirmed that SEA is one of the regions with the most serious biomass combustion and air pollution in the world, especially in Laos and Thailand. These regions distributed numerous active fires and had high concentrations of $PM_{2.5}$ and fire carbon emission (FCE) [5,8]. Almost every January to April, severe particulate matter problems arise in the north of Laos and Thailand with the help of two major contributors: wildfires and agricultural burning [9]. Several studies also mentioned that, in addition to negatively affecting local air quality, the emissions from open-air biomass combustion can have an adverse regional and global impact [10,11]. Therefore, an in-depth study of the relationship between $PM_{2.5}$ and FCE, as well as their long-term distribution, is of great significance for a better understanding of the evolution and origin of $PM_{2.5}$ in the rainforest of the northern tropics. $PM_{2.5}$ can be discharged directly or formed by pollutant precursors in the atmosphere. The latter form of $PM_{2.5}$ is influenced by climate; climate change may, therefore, change the concentration and distribution of $PM_{2.5}$ [12,13].

Vegetation in tropical regions has reduced adaptability to climate change and seasonal conditions relative to vegetation in other regions [6]. Thus, climate change may have greater effects on the FCE and air quality in the tropics than elsewhere. Climate is a major driver of wildfires, subsequently affecting carbon emissions by regulating vegetation productivity and fuel moisture [14]. On the other hand, climate change and FCE significantly exacerbate air pollution and complicate its control [8]. Yet many recent studies focused on the effects of pollutants on local climate (e.g., air temperature, radiation forcing, and precipitation) or the impact of weather conditions on pollutants in the short term [15,16]. Fewer studies have assessed how the long-term climate affected air quality in turn, especially in tropical rainforests that alternate between dry and wet seasons. For instance, $PM_{2.5}$ concentrations in the dry season were higher than in the rainy season due to the reduced precipitation and the increased biomass combustion, which greatly affected the annual average pollutant concentration level [7,17]. It is well known that the increase in carbon emissions aggravates the greenhouse effect and raises the ambient temperature, while the higher temperature is one of the factors affecting the dispersion of $PM_{2.5}$. The complex influence of temperature on $PM_{2.5}$ is an open area of research [18]. Therefore, it is not enough to carry out air quality management without a sufficient understanding of the impact of climate change and FCE on air quality.

Given that the control of open biomass combustion has become an important investment in air quality management, and that the fire risk due to climate change and anthropogenic activity is expected to increase, efficiently identifying the drivers of $PM_{2.5}$ and FCE is becoming increasingly important. In boreal forest, $PM_{2.5}$ produced by fire emissions has been widely studied. A complex feedback circuit has been formed between climate warming and increased lightning, increased fires, and increased pollutant emissions, which has led to increasing forest fires in this area [19,20]. In tropical Southeast Asia, however, deforestation, grazing, and burning have increased fire frequency. Meanwhile, some underdeveloped countries do not have enough funds to support fire prevention infrastructure and equipment; hence, it is difficult to effectively control fire when a fire breaks out, which expands its spread while increasing fire emission [21,22]. However, little remains known about the effects of climate change and human disturbance on $PM_{2.5}$ and carbon emissions arising from fires in tropical regions. To this end, the following objectives are proposed in this research: (1) to understand the spatial distribution of $PM_{2.5}$ and FCE in tropical rainforests of northern hemispheres, (2) to understand the contribution of fire-driven carbon emissions to $PM_{2.5}$ in tropical rainforests of northern hemispheres,

and (3) to identify the drivers of PM$_{2.5}$ and FCE in tropical regions. These explorations and findings will facilitate the understanding of climate and residents' control of PM$_{2.5}$ and FCE in SEA, the establishment of predictive models for PM$_{2.5}$ and FCE, and the improvement of environmental quality in SEA countries.

## 2. Materials and Methods

### 2.1. Study Area

SEA is an important source of global biomass burning (BB) emissions [23], and Laos is one of the major emitters. The study area is located in the tropical rainforest of northern Laos (NL: 17.5°–22.5°N, 100°–105°E) (Figure 1) and has a tropical monsoon climate. The climate is characterized by high annual temperatures (average ~26 °C) with obvious dry and wet seasons in a year. The wet season is from May to October, with the southwest monsoon prevailing, and the dry season is from November to April of the following year, with a prevailing northeast wind. The whole region has abundant annual rainfall, with an annual precipitation of 1250–3750 mm. Due to the humid tropical climate in Southeast Asia, this region has formed highly dense tropical forests. About 80% of the whole region constitutes mountains and plateaus, most of which are covered by forests, and the terrain in the east (Khouang Plateau) is slightly higher than that in the west. According to active fire data and PM$_{2.5}$ data in the period of 2003 to 2019 provided by the National Aeronautics and Space Administration (NASA), we believe that a large number of cases of biomass burning in the entire SEA tropical may have been associated with particulate matter emissions. According to the MODIS fire data, the average number of fires in Laos was 39,666 annually in 2003–2016, most of which were artificial [24].

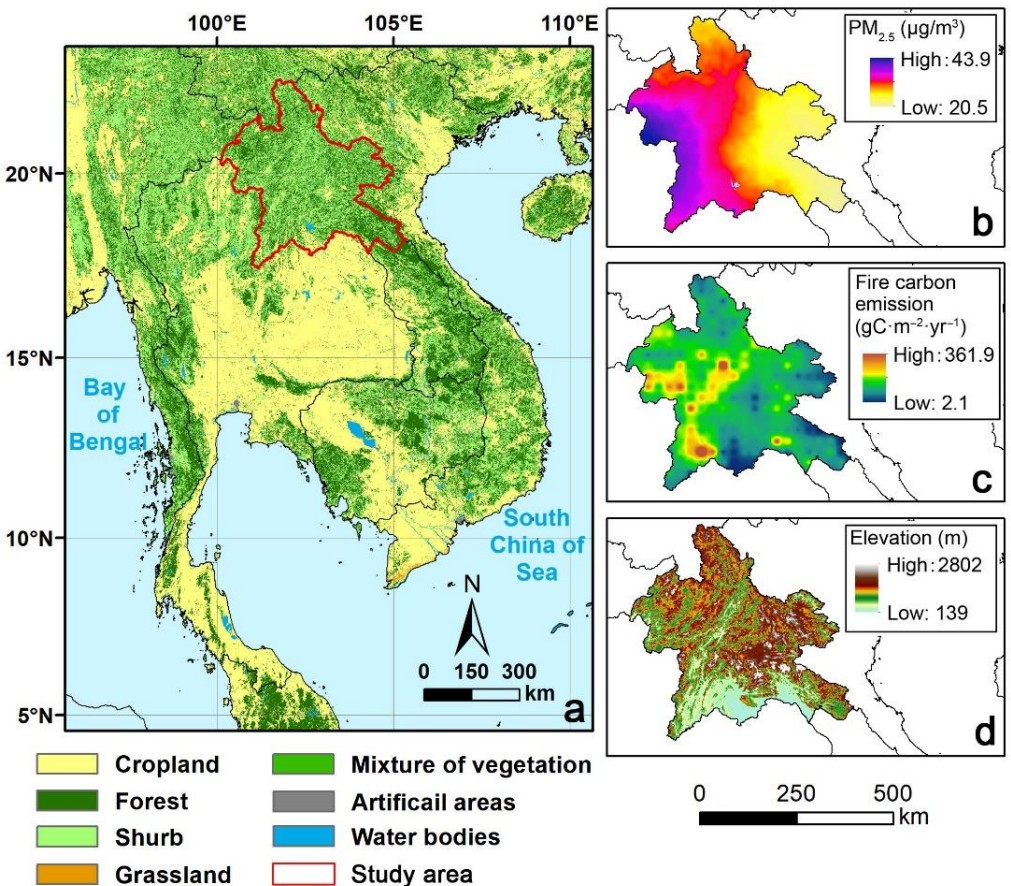

**Figure 1.** Study area. (**a**–**d**) Vegetation types, PM$_{2.5}$ (average from 2003 to 2019), fire carbon emission (average from 2003 to 2019), and altitude distribution map in the study area, respectively.

### 2.2. Data Collection and Process

### 2.2.1. PM$_{2.5}$ Data

The PM$_{2.5}$ data in this study came from the NASA Socioeconomic Data and Applications Center (SEDAC) (https://sedac.ciesin.columbia.edu/ (accessed on 13 March 2022)). These PM$_{2.5}$ estimates were translated from aerosol optical depth (AOD) retrievals obtained by combining the NASA Moderate Resolution Imaging Spectroradiometer Collection 6.1 (MODIS C6.1), Multiangle Imaging SpectroRadiometer Version 23 (MISRv23), MODIS Multi-Angle Implementation of Atmospheric Correction Collection 6 (MAIAC C6), and the SeaViewing Wide Field-of-View Sensor (SeaWiFS) satellite algorithms. After that, the AOD data were correlated with surface PM$_{2.5}$ using the GEOS-Chem chemical transport model and combining geographically weighted regression (GWR) to adjust the PM$_{2.5}$ bias of each pixel in the satellite's initial values [25]. Since this method has been used to create long-term PM$_{2.5}$ data records with relatively fine spatial resolutions (~1–10 km), many scholars have used them to study various spatial scales or long-term regularities [26]. The temporal and spatial resolutions of the global annual PM$_{2.5}$ grid data are annual and 0.01°, respectively.

### 2.2.2. The Fire Carbon Emission (FCE) Data

FCE data (unit: $gCm^{-2} \cdot month^{-1}$) used in the study came from the fourth version of the Global Fire Emissions Database (GFED4s), which combines satellite information on fire activity and vegetation productivity to estimate monthly burn area and fire emissions [5]. GFED4 has been used in large-scale atmospheric and biogeochemical studies [27,28]. The spatial and temporal resolution of these data are 0.25° and monthly, respectively, and they are available from http://www.globalfiredata.org (accessed on 14 March 2022).

### 2.2.3. Climate Data

CRU TS (Climatic Research Unit gridded Time Series) is one of the most widely used meteorological datasets produced by the UK's National Center for Atmospheric Sciences (NCAS) [29]. CRU TS provided our study with monthly mean temperature (TMP; 2 m above the surface), diurnal temperature range (DTR; 2 m above the surface), precipitation, and potential evapotranspiration data with 0.5° resolution from 1901 to 2020 (https://crudata.uea.ac.uk/cru/data/hrg/ (accessed on 17 March 2022)). The availability of these datasets including previous and current versions (updated to Version 4) was discussed in [29,30].

Severe drought and the subsequent increase in wildfires have been confirmed as key processes of climate change [31]. The general definition of drought is based on the aridity index (AI), i.e., the ratio of annual average precipitation (P) to potential evapotranspiration (PET). Therefore, AI was calculated using the precipitation and potential evapotranspiration variables in the above datasets.

The information on soil moisture is the volume of water in the soil surface layer (0–7 cm) provided by the ECMWF Integrated Forecasting System (downloaded from https://cds.climate.copernicus.eu/(accessed on 18 March 2022)). It is a post-processed subset of the full ERA5-Land dataset [32]. The soil moisture content changes are associated with meteorological factors (mainly precipitation), soil characteristics (porosity, weight, permeability, etc.), vegetation conditions, and human activities [33,34]. In turn, the lack of water in the soil leads to insufficient moisture absorption by plant roots and plant transpiration also makes plants lose a lot of water, resulting in plant moisture loss and dryness, which increases the flammability of fuel [34].

### 2.2.4. Vegetation and Topography Data

Leaf area index (LAI), also known as leaf area coefficient, refers to the projected area of leaves over per unit ground surface ($m^2 \cdot m^{-2}$). It plays a vital role in energy (including radiation) and matter exchange between the crown and atmosphere. LAI is an important structural property of vegetation and is used to characterize vegetation productivity and eco-hydrological information. Furthermore, LAI has been identified by researchers as one

of the influencing factors for forest regulation of $PM_{2.5}$ and wildfires [31,35]. LAI datasets used in this study came from the Copernicus Climate Change Service (C3S) Data Platform (https://cds.climate.copernicus.eu/ (accessed on 18 March 2022)).

Elevation data, which can be found at https://pubs.er.usgs.gov/ (accessed on 18 March 2022), were sourced from the Global Multi-resolution Terrain Elevation Data 2010 (GMTED2010) jointly produced by the United States Geological Survey (USGS) and the National Geospatial Intelligence Agency (NGA) [36]. The elevation data are available at three different resolutions (about 1000, 500, and 250 m). We used the highest resolution (i.e., 250 m) for our study.

### 2.2.5. Anthropic Factors

Anthropic factors are very important in wildfire prediction and fire risk assessment. Especially in the tropics, the majority of wildfires are caused by human activities, through accidents, negligence, or intentional fire acts influenced by indigenous fire culture [37]. With the increasing impact of human activities on the tropical rainforest ecosystem, human activities with more complex internal factors challenge the traditional single indicator to describe the impact of human activities. In this study, we used the human footprint and degree of hemeroby (a method to assess the naturalness of vegetation) to characterize and quantify the intensity of human activities according to the research results of Liu et al. [38]. Six spatial data indicators (population density, land use, grazing density, nighttime lighting data, and railway and highway buffer zone) were used to establish the human footprint index of the study area from 2003 to 2019, and the data of the protected areas, lakes, and reservoirs and other points of interest were corrected [39]. The human interference degree is based on the hemeroby scale to evaluate the impact of human activities on the ecosystem by assigning different interference types (land-use types) (see Beyhan et al. [40] for assignment criteria.).

### 2.2.6. Scale of Study Cell

In order to have a unified spatial scale, ArcGIS 10.6 software was adopted to divide the study area into 6690 5 × 5 km grid cells. Then, all data variables were redistributed to these cells. This task was performed using the "zonal statistics as table" tool in ArcGIS. Summary statistics of all variables are given in Table 1.

**Table 1.** Global Moran's *I* of $PM_{2.5}$ concentration and fire carbon emission in the study area.

| Variables | Region | Global Moran's *I* | *z*-Score | *p*-Value |
|---|---|---|---|---|
| $PM_{2.5}$ concentration | NL | 0.996 | 112.664 | <0.0001 |
| Fire carbon emission | NL | 0.963 | 110.963 | <0.0001 |

### *2.3. Data Analysis*
### 2.3.1. Spatial Analysis

We conducted a spatial autocorrelation analysis using ArcGIS 10.6 to explain the spatial correlation among all $PM_{2.5}$ concentrations and carbon emission from wildfire units in the study area. The global Moran's *I*, *z*-score, and *p*-values were used to describe the spatial correlation degree and significance of all cells for $PM_{2.5}$ concentrations and carbon emission in the study area. The global Moran's *I* is defined as follows [41,42]:

$$I = \frac{n}{\sum\limits_{i=1}^{n}\sum\limits_{j=1}^{n} w_{i,j}} \cdot \frac{\sum\limits_{i=1}^{n}\sum\limits_{j=1}^{n} w_{i,j}(X_i - \overline{X})(X_j - \overline{X})}{\sum\limits_{i=1}^{n}(X_i - \overline{X})^2}, \tag{1}$$

where $X_i$ and $X_j$ denote the observed values of the variable under study at locations *i* and *j*, respectively, $\overline{X}$ is the average of $X_i$ over the *n* locations, and $w_{ij}$ is the spatial weight

measured within a given distance or bandwidth. If location $j$ is a neighbor of the subject location $i$, $w_{ij} = 1$; otherwise, $w_{ij} = 0$. A value of $I > 0$ indicates spatial positive correlation, while $I < 0$ indicates a spatial negative correlation, and $I = 0$ reflects that the space is random.

In addition, the *z*-score of global autocorrelation statistical data is calculated as follows:

$$z_I = \frac{I - E[I]}{\sqrt{V[I]}},\tag{2}$$

where $E[I] = -\frac{1}{n-1}$ and $V[I] = E[I^2] - E[I]^2$ represent the expectation and variance of the global Moran's $I$, respectively.

Corresponding to the global Moran's $I$ is the local Moran's $I$. It can be clearly seen that the global Moran's $I$ is the mean of the local Moran's $I$ to $i$ (see Equation (3), where parameter interpretation is the same as for Equation (1)) [42,43]. The local Moran's $I$ reflects the spatial autocorrelation of the units in the study area, i.e., the correlation between a certain spatial unit and its adjacent spatial unit. We used the cluster/outlier analysis (Anselin local Moran's $I$) in ArcGIS 10.6 to understand the spatial correlation patterns of $PM_{2.5}$ concentrations and wildfire carbon emissions in different spatial locations. The tool has the ability to identify spatial clusters of high- or low-value cells and determine spatial outliers (high value surrounded by a low value or low value surrounded by a high value). Clustering/outlier analysis can also yield local Moran's $I$, accompanied by the *z*-score (Equation (4)) and *p*-value to explain the statistical significance of local Moran's $I$.

$$I_i = \frac{n(X_i - \overline{X}) \sum\limits_{j=1, j \neq i}^{n} w_{i,j}(X_j - \overline{X})}{\sum\limits_{i=1}^{n}(X_i - \overline{X})},\tag{3}$$

$$z_{Ii} = \frac{I_i - E[I_i]}{\sqrt{V[I_i]}}\text{m}\tag{4}$$

where $E[I_i] = -\frac{\sum\limits_{j=1, j \neq i}^{n} w_{i,j}}{n-1}$ and $V[I_i] = E[I_i^2] - E[I_i]^2$ represent the expectation and variance of the local Moran's $I$, respectively.

### 2.3.2. Random Forest (RF) Regression

We used an established technique, namely, random forest (RF), to build predictive models and analyze the relationship between predictors and $PM_{2.5}$ and FCE. RF can perform classification or regression predictions depending on whether the target variable type is categorical or continuous [44]. Assuming that the number of samples and variables in the training set is $N$ and $M$, respectively, the algorithm of RF is as follows: (1) $N$ samples are randomly selected from a training set to generate a regression tree; (2) randomly selected m ($<M$) variables are selected on each node as candidate variables to split this node. The number of variables on each node needs to be consistent; (3) the results of each regression tree are integrated to generate the predicted values; (4) the response variables corresponding to the sample points that are not used when generating the tree can be estimated by the generated tree, and the out-of-bag (OOB) error can be obtained by comparing with the true value [45]. This is a major advantage of RF when test sets are not available. In addition, the RF algorithm can calculate the relative importance of explanatory variables. These reasons led us to consider RF to perform the statistical analysis of this study. It has the highest accuracy among all the similar algorithms (decision tree) at present and can effectively process the data with many missing data [45,46]. This is why RF has been widely used in past studies to identify factors contributing to wildfires and air pollution [47,48].

It is worth noting that two important parameters, i.e., the number of regression trees (*ntree*) and the number of optimal competitive variables of nodes (*mtry*), need to be determined in the calculation of RF. With the increase in the number of regression trees,

the error of a random forest will decrease until it tends to be stable, signifying an optimal value. The parameter *ntree* is calculated after determining *mtry*. When the errors in the model are stable, the minimum value of *ntree* is used as the parameter to train the model (the calculation result is shown in Result.). For the value of *mtry*, we refer to the suggestion of Liaw and Wiener [49] to select *mtry* = $M/3$ for RF regression, where $M$ is the number of variables.

### 2.3.3. Assessment of Variable Importance

By using the variable importance measure of the random forest algorithm, the importance of features can be ranked, and the features with higher importance can be selected. In the RF model, the common evaluation indicators of variable importance measurement are the increase in mean square error (IncMSE) and the increase in node purity (IncNodePurity). IncMSE refers to the increase in the error estimated by the RF model relative to the original error after the random value of the variable. A larger IncMSE value indicates a more important variable. IncNodePurity refers to the influence degree of the variable on each decision tree node. A greater value indicates a more important variable, while a lower value indicates relative unimportance. In this study, IncNodePurity was used as the evaluation index of variable importance.

### 2.3.4. Evaluation of RF

Several parameters that have been widely used in previous studies were considered as model evaluation indicators in this study. The coefficient of determination *R*-squared was used to describe the fitting effect between observed and RF algorithm estimated response variables. Mean absolute error (MAE) is the average of the absolute value of the error between the predicted value and the real value. It was used to describe the error between the observed and predicted fire density. Root-mean-square error (RMSE) was also used to measure the error rate of the regression model, which can detect the dispersion of error. However, RMSE could be dominated by some large values to bring deviation to many values because of the wide range of predicted values in our research. In order to solve this problem of RMSE evaluation, root-mean-square logarithmic error (RMSLE), whereby the predicted and the observed values were logarithmized before calculating the RMSE, was utilized. In addition, RF technology provides users with an index (percentage of the variable explained, % Var.explained) of the explanatory ability of the model composed of predictive variables to the changes of target variables.

In this study, we carried out RF regression analysis in R environment software through the "*rfPermute*" package, which can calculate the significance of predictors' importance. Meanwhile, the significance of the model was evaluated using the "*A3*" package in R [50].

### 2.3.5. Structural Equation Model (SEM)

The structural equation model was used to test the direct and indirect effects of potential factors on PM$_{2.5}$ and FCE. Structural equation modeling (SEM) is a method of establishing, estimating, and testing causal relationship models among variables, also known as covariance structure analysis. This method is a combination of factor analysis and multiple regression analysis, which is used to analyze the structural relationship between response variables and predictors. Due to the ability to estimate the dependencies among multiple interrelated variables in one analysis [51], SEM has gradually become the highlight of articles in various fields, serving for the systematic summary and analysis of the ideas and results of articles [51,52]. In this study, the modeling of the path and the structural relationship of variables were realized using the "*lavaan*" package in R. According to previous studies, the validity and parsimony of the model were judged using the root-mean-square error of approximation index (RMSEA < 0.05), standardized root-mean-square residual index (SRMR < 0.08), goodness-of-fit index (GFI > 0.90), and comparative fit index (CFI > 0.90) [52].

## 3. Results

### 3.1. Spatial and Temporal Distribution of FCE and PM$_{2.5}$

The results of the global spatial autocorrelation analysis show that the global Moran's *I* values of the study areas were greater than 0, and all of them passed the 5% significance level test (Table 2), which indicates that the spatial correlation of FCE and PM$_{2.5}$ in the study area was positive. In order to further explore the spatial association, difference degree, and aggregation distribution between local and surrounding areas, the research team obtained the local Moran's *I* scatter diagram, in which the spatial patterns of PM$_{2.5}$ exposure and FCE in study regions were analyzed using the local spatial statistical technique. The high/high cluster (HH in Figure 2a,b) indicated that the values of PM$_{2.5}$ exposure and FCE of this area were higher than the average values of the whole region, and the values of region around the high PM$_{2.5}$ exposure and FCE were also higher than the average values, which reflected the positive autocorrelation of the variable. Similarly, the low/low cluster exhibited that the values of PM$_{2.5}$ exposure and FCE were lower than the average values of the whole region, and the values of their surrounding area were also lower than the average values, which was also a positive autocorrelation. Negative autocorrelation included low/high (LH) and high/low (HL) outliers; however, no negative spatial autocorrelation was found in our study. According to the spatial clustering degree of PM$_{2.5}$ concentration, the number of HH cells was higher than that of LL cells (52.8% > 46.5%, Figure 2a) in NL, but the FCE was the opposite (HH: 42.3% < LL: 55.7%, Figure 2b).

In view of the fact that the scatter plot cannot intuitively identify the local correlation types and clusters and their statistical significance, we drew the local indicators spatial autocorrelation (LISA) cluster map (Figure 2c–f). A LISA diagram is mainly used to test the local autocorrelation of variables. In terms of the aggregation effect for both PM$_{2.5}$ concentration and FCE, the areas with statistical significance were greater than those without significance in NL. In addition, most of the significant areas belonged to an extremely significant level ($p < 0.001$) (Figure 2d,f). As expected, the LISA diagrams indicated that not all types of clustered areas were significant. Taking the distribution of PM$_{2.5}$ exposure as an example, the global Moran's *I* scatterplot illustrates that the HH regions covered 42.3% of cells, but only 22% of the statistically significant areas.

**Table 2.** The summary statistics of the variables for the studied region. Note: FCE is the fire carbon emission, AI is the aridity index, TMP is the annual average temperature at 2 m, DTR is the average daily temperature range, LAI is the vegetation leaf area index, and SM is the soil moisture.

| Variables/Unit | Minimum | 1st Quartile | Median | Mean | 3rd Quartile | Maximum |
|---|---|---|---|---|---|---|
| PM$_{2.5}$ (ug·m$^{-3}$) | 21.35 | 26.52 | 29.61 | 29.26 | 31.75 | 37.56 |
| FCE (PgC·year$^{-1}$) | 2.331 | 52.926 | 69.417 | 73.9 | 89.581 | 285.239 |
| AI | 1 | 1.265 | 1.354 | 1.402 | 1.545 | 1.898 |
| TMP (°C) | 20.29 | 22.38 | 23.01 | 23.16 | 23.76 | 27.04 |
| DTR (°C) | 7.099 | 8.571 | 9.373 | 9.501 | 10.476 | 11.757 |
| Elevation (m) | 153.5 | 579.5 | 799.8 | 803.8 | 1025.2 | 2234.2 |
| Footprint | 0.0951 | 10.4115 | 15.696 | 16.0133 | 20.4711 | 57.8405 |
| Hemeroby | 0.7017 | 1.4616 | 1.7478 | 1.8271 | 1.9724 | 6.4505 |
| LAI | 54.07 | 374.21 | 450.49 | 444.06 | 524.3 | 814.82 |
| SM | 0.274 | 0.3969 | 0.407 | 0.4046 | 0.4171 | 0.4512 |

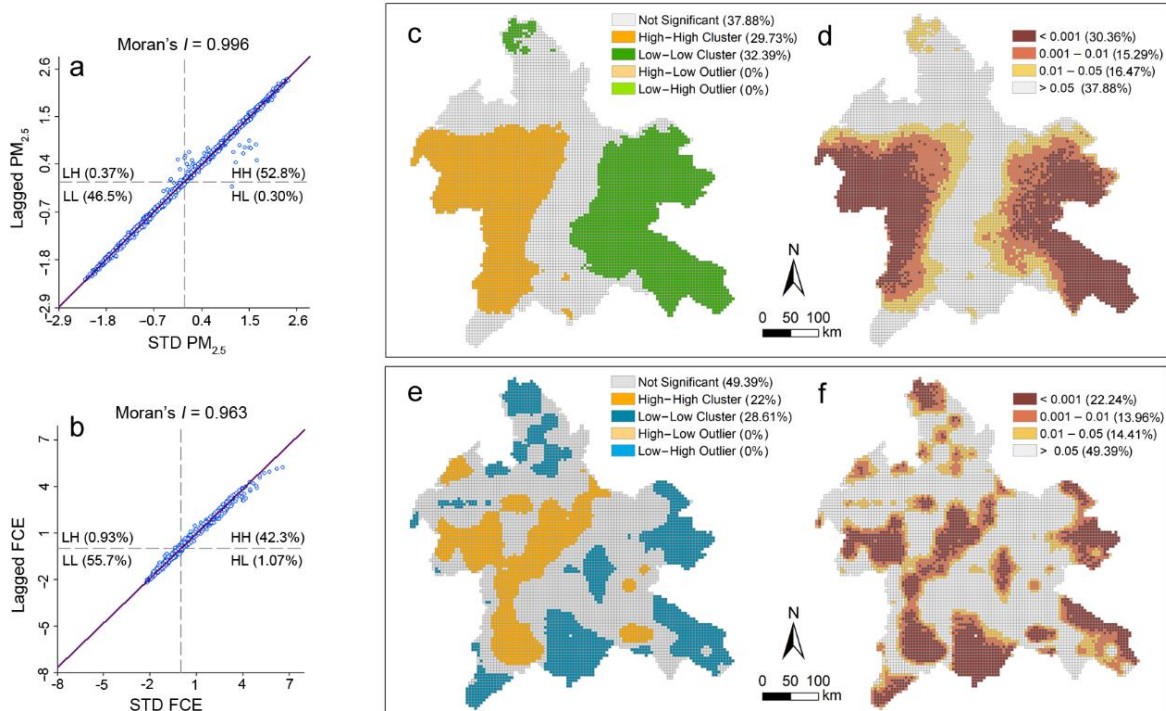

**Figure 2.** Spatial distribution pattern of PM$_{2.5}$ concentration and FCE in the study area based on cluster and outlier analysis (Ansel in local Moran's *I*). (**a**,**b**) Moran's *I* scatter plot of PM$_{2.5}$ concentration and FCE in NL, respectively. The abscissa is the observed value of PM$_{2.5}$ concentration and FCE of a spatial unit (after standardization), while the ordinate is the "lagged" value of the spatial unit, i.e., the average value of the observed PM$_{2.5}$ concentration and FCE of adjacent units (after standardization); (**c**,**e**) LISA agglomeration of PM$_{2.5}$ concentration and FCE in study areas; (**d**,**f**) distribution of areas significant or not for LISA agglomeration.

### 3.2. The Importance of Influencing Factors of PM$_{2.5}$ and FCE

We employed RFs to describe the relationship between PM$_{2.5}$ exposure and FCE, as well as their drivers. The results reveal that the fitting of the RF was statistically significant (*p*-value < 0.001 in Figure 3a,b). Figure 4 demonstrates the importance of explanatory variables affecting PM$_{2.5}$ and FCE during the entire 17 year period through RF (17 year average). From the results, it is clear that climate factors had the most important impact on PM$_{2.5}$ concentration and FCE in all variables, especially AI and DTR. Among the explanatory variables of PM$_{2.5}$ concentration, AI was the most important factor, followed by the effect of DTR. Differently, DTR was the most important followed by AI in explanatory variables of FCE. Additionally, the importance of TMP to PM$_{2.5}$ and FCE in NL was also very significant. Another variable worthy of attention was SM, which had no significant importance to PM$_{2.5}$, but it had outstanding performance among many factors of FCE. At the same time, it could be known through the local importance diagram that variables important in the global ranking were also of the most importance in the local importance ranking, such as AI and DTR (Figure 3c,d).

In addition, our findings on annual analysis imply that climatological factors maintained high importance among the factors affecting PM$_{2.5}$ concentration and FCE each year, while human impacts were weak and stable for both PM$_{2.5}$ and FCE. In NL, the order of factors was relatively stable each year; AI was the most important influence factor of PM$_{2.5}$ concentration in each year, followed by DTR and TMP, which was consistent with the average result of 17 years (Figure 4a). Similarly, the climate was the dominant influencing factor of FCE. However, AI's contribution to FCE was not always the strongest over the period 2003–2019. The most important factors affecting FCE alternated between AI and DTR (Figure 4b).

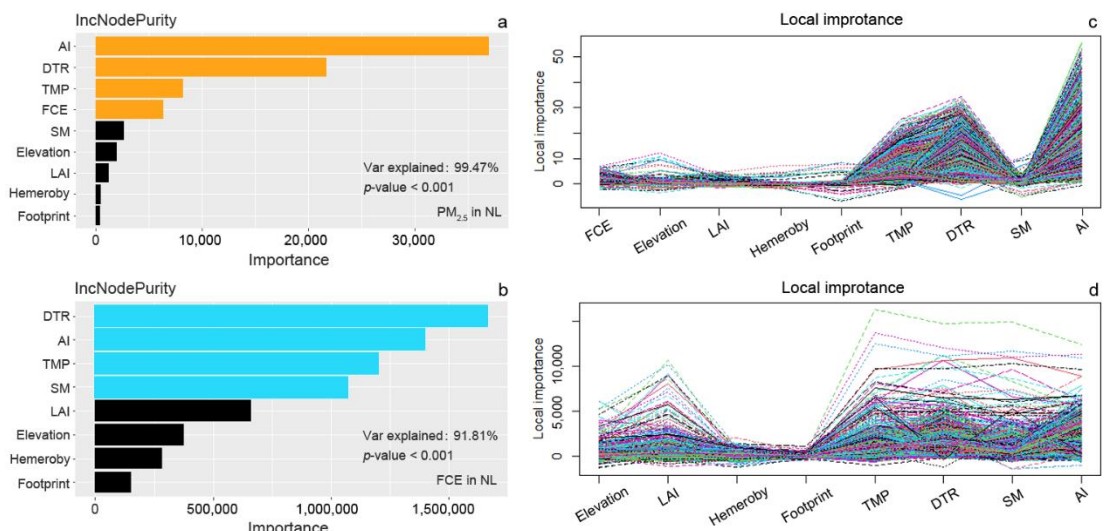

**Figure 3.** Variable importance rank (IncNodePurity—increase in node purity) of $PM_{2.5}$ concentration (**a**) and FCE (**b**) from RF in NL in 17 years (average). The orange and blue bars represent the significant ($\alpha < 0.05$) importance of the variable, while the black bar represents the insignificant importance of the variable. Var explained reflects the overall explanatory rate of predicted variables to $PM_{2.5}$ and FCE related to variance; *p*-value is the significance of the RF model based on complete variables. (**c**,**d**) Local importance maps of the $PM_{2.5}$ concentration and FCE variables, respectively.

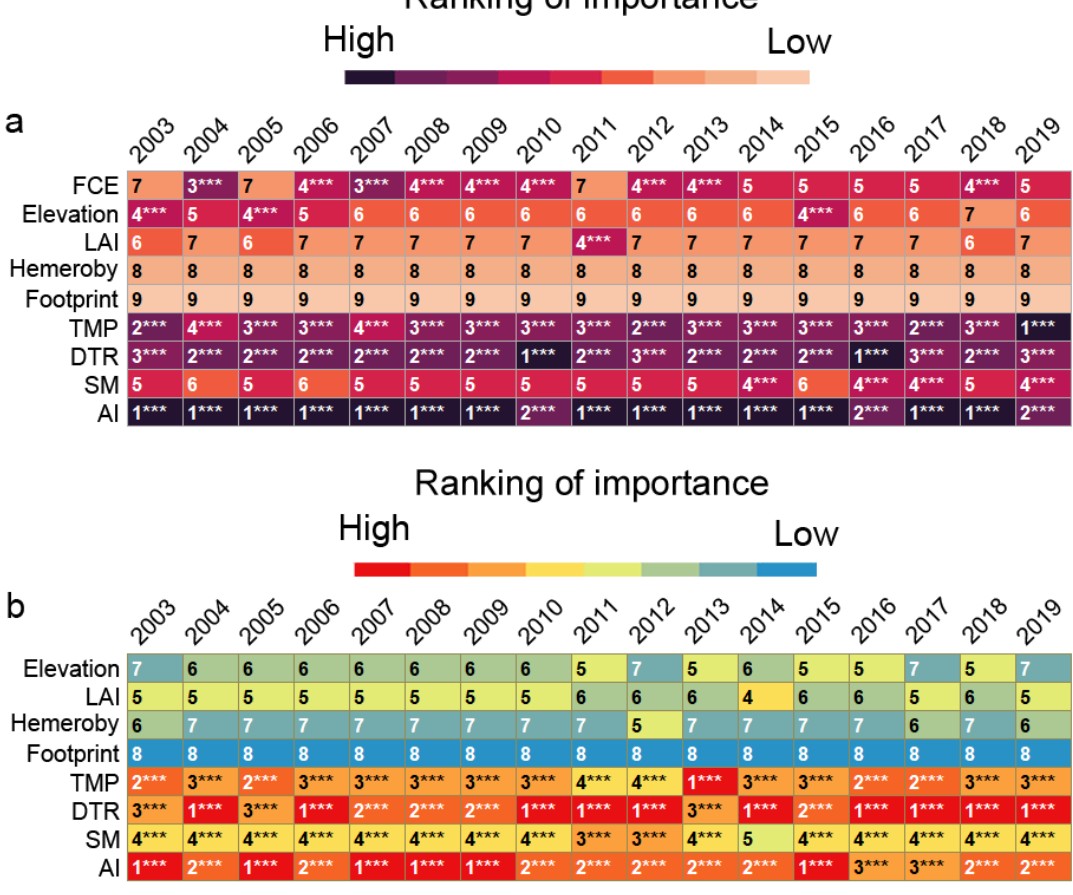

**Figure 4.** The effect of annual variables by RF on $PM_{2.5}$ concentration and FCE in NL. (**a**,**b**) Ranking of importance of variables affecting $PM_{2.5}$ concentrations and FCE in NL, respectively. "***" represents significant at 1% level, that is, $p < 0.01$.

### 3.3. The Variables Selected and Goodness of Fit for RF

According to the characteristic that the error of RF model would decrease with the increase in the number of regression trees and the number of variables, we calculated the relationship between the number of trees and the model error (Figure 5a,b), and then the relationship between the number of variables and the error was verified by 10-fold cross-validation (Figure 5c,d). The results show that, when the number of trees was greater than 500, the error rate of the model was basically stable and consistent for both PM$_{2.5}$ and FCE fitting. For the data of this study, 500 trees were enough to stabilize the error but not too many so as to lead to overfitting. In order to shorten the running time of the RF model, we reduced the number of trees under the condition of ensuring effectiveness; we set the parameter *ntree* = 500 for the RF operation.

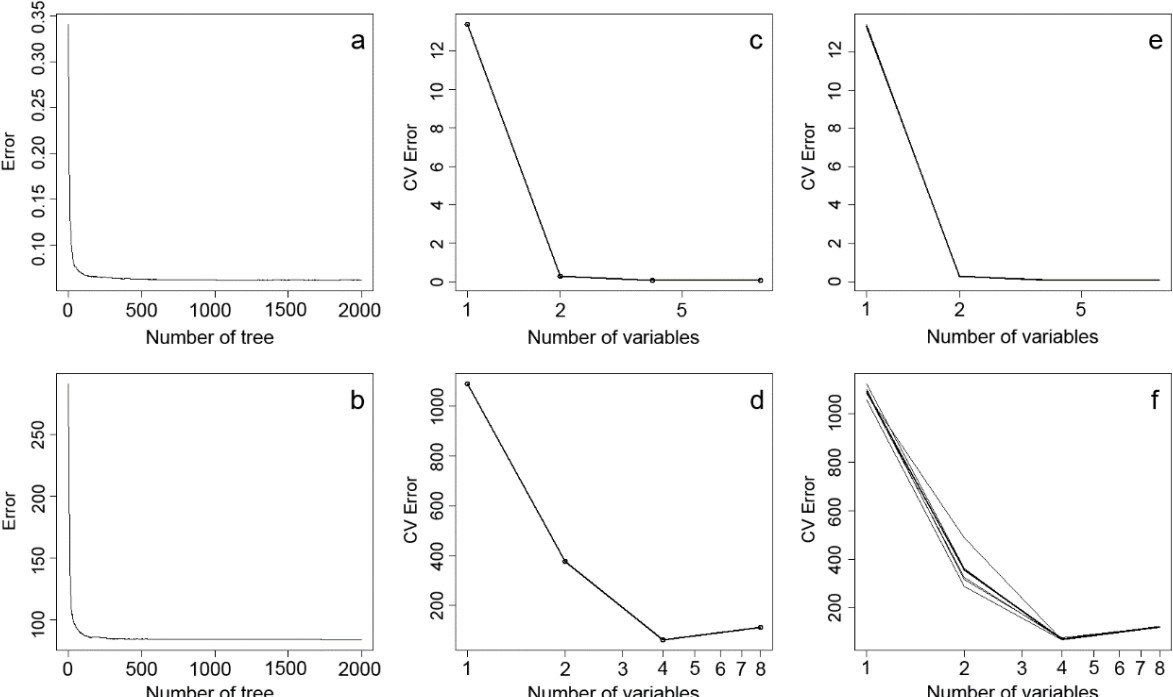

**Figure 5.** Random forest error graphs. (**a**,**b**) Relationship between the number of trees and the error of the RF model for PM$_{2.5}$ and FCE, respectively; (**c**,**d**) relationship between the number of variables and the error of the RF model for PM$_{2.5}$ and FCE under 10-fold cross-validation, respectively; (**e**,**f**) relationship between the number of variables and error of RF model for PM$_{2.5}$ and FCE under multiple cross-verifications, respectively.

Figure 5c,d reveal that the number of optimal variables was 4 for PM$_{2.5}$ and FCE. Furthermore, Figure 5e,f compare the replicate cross-validation curves of PM$_{2.5}$ and FCE fitting with environmental factors, which shows that the calculation results were reliable, especially the fitting results of PM$_{2.5}$ and explanatory variables. It is worth mentioning that the order in which variables changed in Figure 5c–f was determined by their importance, i.e., the first four variables in Figure 3a,b were determined to be the optimal variables (i.e., PM$_{2.5}$: AI, DTR, TMP, FCE; FCE: DTR, AI, TMP, SM) on the basis of the number of variables corresponding to the minimum error. This finding was consistent with the results at the significance level.

Figure 6 displays the relationship between the observed and predicted values of PM$_{2.5}$ and FCE generated by RF. The $R^2$ values of the regression fitted line between the observed and predicted PM$_{2.5}$ exposure and FCE in the study region exceeded 0.98. Moreover, there was not much difference between the RMSEs in the fitting results of the complete and optimal variable for the fit of the PM$_{2.5}$ or FCE, and MAEs were no exception. Meanwhile, we also found that the variance explained in the RF fitting of PM$_{2.5}$ and environmental fac-

tors was higher than the fitting result of FCE and environmental factors (99.47% > 91.81%) (Figure 3). All evidence suggested that RF had an excellent ability to simulate the relationship among PM$_{2.5}$ concentration, FCE, and environmental factors in NL.

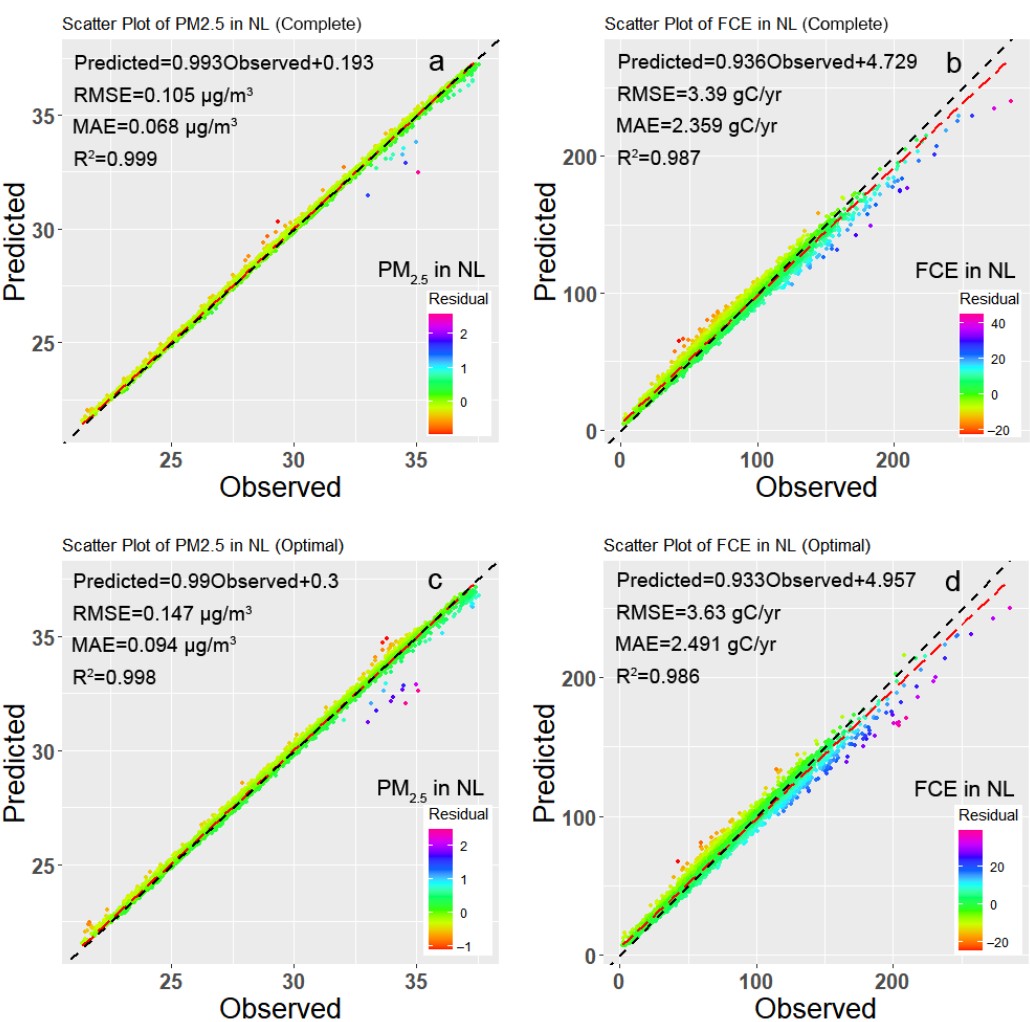

**Figure 6.** Scatterplots of the observed and predicted PM$_{2.5}$ and FCE from the RF regression (**a**,**b**) complete and (**c**,**d**) optimal variables during 2003–2019.

### 3.4. Climate Factors Control on PM$_{2.5}$ Exposure and FCE

Superior results were seen for the utility of SEM-based on variables selected by RF in the study area. The fitting indices of the model were all within the range of the specified values (GIF = 1 > 0.9, CIF = 0.999 > 0.9, RMSEA = 0.045 < 0.05, SRMR = 0.004 < 0.08). The main results of the relationship path diagram given by SEM were consistent with RF. The meteorological factors had dominant control over PM$_{2.5}$ and FCE under standardized data, but the intensity of the relationships between them was slightly different (Figure 7a). AI had the strongest direct influence on PM$_{2.5}$ exposure, followed by DTR and FCE (their normalized coefficients were 0.32 and 0.22, respectively), while the weakest direct influence was seen for TMP (Figure 7b). This suggested that AI had the highest contribution of 0.5 to PM$_{2.5}$, while TMP had the weakest contribution of 0.17.

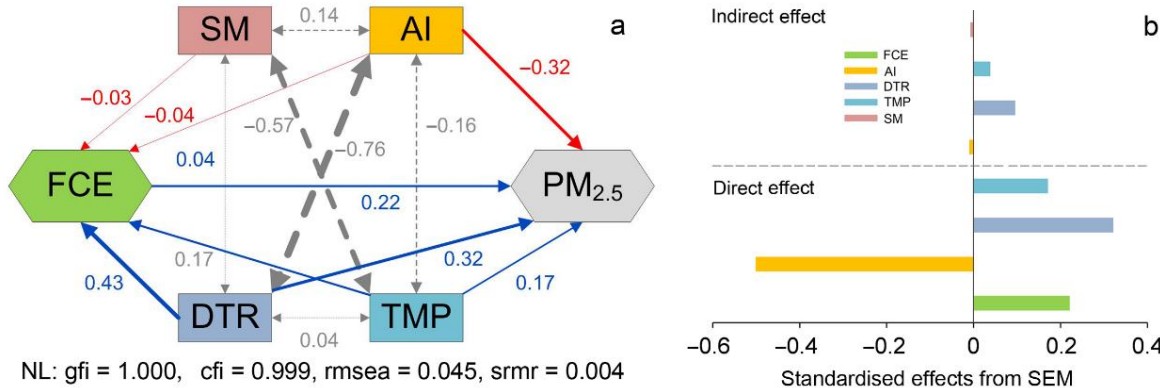

**Figure 7.** The impacts of meteorological factors on PM$_{2.5}$ exposure and FCE in NL as estimated using structural equation modeling (**a**). Blue and red lines indicate the paths of positive and negative relationships, respectively. The width of the lines represents the strength of standardized path coefficients. Gray dotted lines denote the covariance of exogenous variables. Single-head arrows present the hypothesized direction of causation. The number beside the dotted line is the standardized path coefficient. (**b**) Direct and indirect influence of climate and FCE on PM$_{2.5}$ exposure. The *x*-axis represents the standardized effects from SEM. Among them, the direct effect value was directly obtained from Figure 7a, and the indirect effect was obtained by multiplying two direct effects.

In addition, the indirect effect of the climatological factors on PM$_{2.5}$ is displayed in Figure 7b. Among them, the strongest indirect effect on PM$_{2.5}$ was seen for DTR (the indirect effect was obtained by multiplying the direct effect of DTR on FCE with the direct effect of FCE on PM$_{2.5}$, i.e., $0.43 \times 0.22 = 0.0946$), followed by TMP ($0.0.17 \times 0.22 = 0.0374$). Simultaneously, the weak indirect effects of AI and SM on PM$_{2.5}$ could be observed in NL. We observed that almost all variables had positive effects on the PM$_{2.5}$ exposure of NL besides AI. On the other hand, the direct influence of DTR on the FCE in NL was much greater than that of the other factors, while TMP had the same effect on FCE as PM$_{2.5}$. Moreover, the direct influence of AI and SM on FCE in NL was only 0.04 and 0.03, respectively.

## 4. Discussion

The spatial autocorrelation technique was used to analyze the spatiotemporal distribution of PM$_{2.5}$ concentrations and FCE in the tropical rainforests of northern Laos from 2003 to 2019. According to the results of spatial cluster analysis, PM$_{2.5}$ in northern Laos showed a significant spatial clustering state. The HH and LL clustering of PM$_{2.5}$ was distributed in the west and east of the study area, respectively. The HH clustering was also found in the west of the study area similar to the FCE distribution, while its LL clustering distribution was scattered. Obviously, the HH cluster distribution area of PM$_{2.5}$ concentration overlapped with that of FCE in a large range (Figure 2). PM$_{2.5}$ and FCE in the west of the study area both exceeded the average level of the whole region, and the adjacent surrounding areas also presented the same situation. This finding is consistent with some previous research results [7–9]. From the perspective of space, the wet season ensures that areas with a large number of forests and shrubs (Figure 1a) have sufficient live or dead combustibles every year, which provides an opportunity for fires to break out in the following dry season. In addition, we found evidence of their consistent distribution in remote sensing data, including the distribution of ignition points in MODIS fire activity products, as well as estimates of FCE and PM$_{2.5}$ concentration data (Figure 1b,c). From a temporal point of view, March–April is the last period of the dry season in the study area, while rising temperatures bring high-frequency wildfires with the help of a large amount of fuel accumulated in the previous wet season [21]. On the other hand, exotic PM$_{2.5}$ may also be one of the reasons for the stable high concentration of PM$_{2.5}$ in the region. Ma et al. [7] mentioned in their study that, under the influence of the perennial

South Asian monsoon, the air mass carried a large amount of flue gas particulate matter from the southwest (India). On the other hand, this gas particulate matter was blocked by the higher terrain in the east (Figure 1d). Nguyen et al. [8] confirmed that the future increase in $PM_{2.5}$ on the Southeast Asian continent will also be related to the increase in pollutant emissions in India, Cambodia, Laos, Thailand, and Vietnam.

Climate factors play a vital role in influencing $PM_{2.5}$ concentrations in northern Laos, and several studies have provided evidence to support this finding [9,17,22,53]. In the present study, we characterized that drought has a strong negative association with $PM_{2.5}$ concentrations and FCE. It is well known that vegetation can absorb pollutants from the air through stomas when there is adequate moisture in the environment. However, in a drought environment, vegetation shrinks stomas to prevent moisture loss, thereby reducing the ability to absorb and capture $PM_{2.5}$ in the air [54]. On the other hand, drought dehydrates the protoplasm in the tree, makes the leaf smaller, and promotes aging while the stomas are closed, resulting in the leaves falling off [55]. This situation reduces the overall wind-proof capacity of the forest and is not conducive to the deposition of particulate matter. However, other studies discovered that the effects of drought on air quality are not limited to vegetation, especially $PM_{2.5}$ and $O_3$. Wang et al. [56] believed that the elevation of $PM_{2.5}$ was attributed to the comprehensive effects of drought on natural emissions (wildfire emission, volatile organic compounds, and dust), as well as their deposition and chemistry. Demetillo et al. [57] analyzed the data of satellites and remote sensing and found that drought caused difficulties in purifying air pollution in California with frequent wildfires. Meanwhile, the researchers also identified the important role of drought factors in the future modeling of air pollution.

We did not find provisionally direct evidence for the effects of diurnal temperature range on $PM_{2.5}$ from the existing literature. We consider the annual average DTR as a prediction index and analyze its impact on $PM_{2.5}$ in this study. The main reason is that DTR related to the maximum and minimum air temperatures is an important indicator of climate change, compared with the annual average temperature, as it contains more information and better reflects the characteristics of regional temperature variation and its impact [21]. We observed a significantly positive effect of annual average DTR on $PM_{2.5}$. This result supports discoveries in other regions, i.e., that the upward trend of changes in maximum and minimum temperatures in recent years is linked to air quality and visibility [58]. One reason is that a larger diurnal temperature range produces a stronger thermal inversion phenomenon. The emergence of the thermal inversion layer hinders the vertical flow of air, which poses a difficulty for the transport and diffusion of pollutants in the atmosphere near the ground [59]. Furthermore, the diurnal temperature range is the main factor influencing wildfires. A larger diurnal temperature range results in a larger solar elevation angle and less resistance to solar radiation caused by the shorter path. This situation reduces the fuel moisture content [21], which increases the frequency of wildfires and subsequent FCE.

It is easy to find the negative relationship between air pollution emission concentration and temperature in the short term from previous research. Indeed, the increase in temperature results in the mixed layer continuously increasing, while the atmosphere diffuses easily in the longitudinal direction. The mixing layer height is an important climate factor that affects the vertical diffusion of atmospheric pollutants [60,61]. In addition, the surface temperature has a greater impact on the height of the mixed layer, and a higher temperature results in a higher mixed layer [62]. Meanwhile, the large thickness of the mixed layer also gives more room for diluting particulate matter [63]. We found, however, a positive effect of annual mean temperature on $PM_{2.5}$ concentration in our study. The fifth assessment report (AR5) of the Intergovernmental Panel on Climate Change (IPCC) noted that seasonal average and annual average temperatures in the tropics were expected to increase more than in the middle latitudes, meaning that Southeast Asia might be more vulnerable to global warming than the rest of Asia. Warming increases the frequency of wildfires, causing a sharp increase in subsequent FCE, while also exacerbating particulate emissions in the air. On the other hand, a possible reason is that the rapid warming of the

near-surface atmosphere leads to the enhancement of the thermal stability of the lower atmosphere under the background of global warming caused by greenhouse gases. This situation further inhibits the vertical movement of the air and worsens the conditions for the dispersion of particulate matter [64,65]. Another reason is that the reduction in ice in the Arctic sea caused by global warming affects the entire atmospheric circulation, especially the more pronounced amplitude of warming in winter. Simultaneously, the weakening of winter winds in East Asia has led to a weakening of the activity of cold air in the north, i.e., the north winds that can blow away smog in the near-surface and upper atmosphere have become weaker and smaller, which provides an opportunity for particulate matter retention [66].

In this study, we observed weak negative feedback of soil moisture on FCE. The water vapor exchange between surface fuel and soil surface regulates the fuel moisture content, which plays a crucial role in the occurrence of forest fires [67]. In the study of peat fire in Indonesia, Dadap et al. [68] found that the drought of soil led to wildfires, and subsequent uncontrolled fires were associated with haze weather in the local and surrounding areas or nations.

*Limitations*

Nevertheless, there were still some limitations in our research. First of all, climate warming increases the probability of pests and also salt soil because of the rise in sea level, affecting rice production in Southeast Asia. In order to make a living, indigenous peoples develop and expand the available land through deforestation and burning, which not only leads to a reduction in forest vegetation, but also causes serious smog problems [69]. However, the human footprint index and the human disturbance index that we considered in the study did not seem to play an important role as representatives of human activities. The calculation of these two indices may have covered up the information of some human activity variables, such that the influence of human activity factors on $PM_{2.5}$ and FCE did not stand out. How to measure the impact of human activities on environmental pollutants needs to be explored in future research. Secondly, socioeconomic factors such as GDP and the release and implementation of environmental protection policies were not considered in this study but will be addressed in the future. Satellite data modeling showed that FCE has a certain impact on $PM_{2.5}$ concentration, and deciphering the impact mechanism represents our next task. In addition, the $PM_{2.5}$ data in this study were from a single source, and the results were limited by the resolution. The current $PM_{2.5}$ datasets have the characteristics of multiple sources; thus, it is also one of the tasks of future research to compare the influences of different $PM_{2.5}$ data sources and their driving factors, especially the comparison and confirmation between satellite data and actual investigation and experimental results.

## 5. Conclusions

In this study, a database of FCE, $PM_{2.5}$, and environmental factors in Laos was established by acquiring satellite grid data with ArcGIS software. We considered spatial autocorrelation techniques to demonstrate the spatial pattern of $PM_{2.5}$ and FCE in northern Laos. on the relationships between the important variables selected by RF and $PM_{2.5}$ and FCE were explored via SEM. The results allow drawing the following conclusions: (1) the concentration of $PM_{2.5}$ and FCE from 2003 to 2019 in the west of the study area was higher than the average value of the whole region. These areas were surrounded by $PM_{2.5}$ and FCE higher than the average value of the whole region. Their spatial distributions were almost consistent, and these areas will become the focus of work related to pollution control in the future; (2) the quantitative analysis gave us a result of a weak but significant impact of fire emissions on $PM_{2.5}$ in the Laotian rainforest; (3) climate factors played a leading role as drivers of $PM_{2.5}$ and FCE in northern Laos from 2003 to 2019. Among them, drought and annual average diurnal temperature range had the greatest impact. In the context of global climate change, this discovery can help regulators and decision-makers better incorporate drought and diurnal temperature range into FCE and $PM_{2.5}$ estimation models.

We suggest, therefore, that the related government and indigenous people should pay attention to reducing PM$_{2.5}$ pollutants caused by biomass combustion in formulating future measures to prevent and control air pollution, especially in an increasing drought and warming environment. In particular, in the improvement of wildfire management and prescribed burning in the dry season, the arid and warming environment should be given enough attention.

**Author Contributions:** Conceptualization, Z.S. and Y.C.; methodology, Z.S. and Z.X.; validation, Y.C. and L.L.; investigation, S.L. and S.W.; data curation, Z.S. and S.L.; writing—original draft preparation, Z.S.; writing—review and editing, Y.C., H.H., and L.L.; supervision, S.L. and Z.X.; funding acquisition, S.W. and S.L. All authors have read and agreed to the published version of the manuscript.

**Funding:** This research was funded by the Natural Science Foundation of Guangdong Province, China (No. 2021A1515010946), the Forestry Science and Technology Innovation of Guangdong Province, China (No. 2020KJCX003), and the Open Foundation of Zhangzhou Food Industry Technology Research Institute (No. ZSY2020203).

**Data Availability Statement:** The data and statistical analysis methods are available upon request from the corresponding author.

**Acknowledgments:** We would like to thank the editor and anonymous reviewers for their useful advice that helped to improve the manuscript.

**Conflicts of Interest:** The authors declare no conflict of interest. The funders had no role in the design of the study; in the collection, analyses, or interpretation of data; in the writing of the manuscript; or in the decision to publish the results.

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
