# Peer review of "Exploration of the Contribution of Fire Carbon Emissions to PM2.5 and Their Influencing Factors in Laotian Tropical Rainforests"

_remotesensing, doi:10.3390/rs14164052_

Round 1

Reviewer 1 Report

The approached subject, is actual and useful for prevent and control of open biomass combustion.

 The obtained results are suggestive represented by tables and figures. Their interpretations are completed and reported to the results available and published in the references. The results are interesting and demonstrate that the  annual analysis imply that climatology factors have maintained high importance among the factors affecting PM2.5 concentration and FCE  each year, while human impacts were weak and stable for both PM2.5 and FCE.  

  The figures are suggestive and show that the climate factors played a leading role in the drivers of PM2.5 and FCE in northern Laos from 2003 to 2019.

 There are some limitations in the study but I propose to the authors to extend it in another work by comparison with actual results.

Reviewer 2 Report

The paper considers fire carbon emission and influencing factors to demonstrate PM2.5 concentrations in Laos. It looks a nice work and timely required research. However, I have few comments in order to improve the paper, as such:

1. The introduction looks consistent. However, link 66, all of a sudden the adaptability of climate change in introduced. In the previous paragraph, a connection should be made to tell the readers about the connection of pm2.5 and climatic changes. Need few citation in this regard as well to open up the discussion in next paragraph!

2. Is there a difference of fire characteristics and propagation in the tropical countries and countries in the North (e.g., Canada, USA)? Authors should emphasize the previous works conducted in the North and demonstrate the difference of their work in the last paragraph in the Introduction. This will enhance the readability of the work.

3. How many fires are happened yearly in Laos and what are the reasons (i.e., natural/man made)? A quick note could be good to know about this in the study area!

4. Data sources look interesting. Method section looks good. A quick note about MODIS data....like spatial resolution. It is considered as a lower spatial resolution data and why have you not performed any fusion technique to enhance the resolution? Although the forested data at 500m resolution should be good but the authors may write the possible limitations in obtaining other data sources if there any.

5. Line 514 in the discussion section, authors should mention about the previous research with citation. Otherwise this sentence looks vague.

6. Conclusion looks nice

Reviewer 3 Report

Line 18. Define the acronym HH.

Line 116. The colour maps are not described in the figure’s caption.

Line 344. Please replace with “studied region”. Please also define all the parameter acronyms in the caption of the Table.

Line 391. Could you better explain why you selected ntree = 500?

Line 436-437. I suggest the Authors to rephrase this sentence because it is unclear in my opinion.

Lines 445-451. It is very difficult to understand the parameter reported on the x-axis in Figure 7b.

I suggest a full linguistic revision of the manuscript.

Author Response

This manuscript is a resubmission of an earlier submission. The following is a list of the peer review reports and author responses from that submission.